# Characterisation of Curing of Vinyl Ester Resin in an Industrial Pultrusion Process: Influence of Die Temperature

**DOI:** 10.3390/polym15183808

**Published:** 2023-09-18

**Authors:** Sai Ajay Chandra Chaparala, Omar Alajarmeh, Tristan Shelley, Xuesen Zeng, Kendric Rendle-Short, Dean Voice, Peter Schubel

**Affiliations:** 1Centre for Future Materials, University of Southern Queensland, Toowoomba, QLD 4350, Australia; saiajaychandra.chaparala@usq.edu.au (S.A.C.C.); tristan.shelley@usq.edu.au (T.S.); xuesen.zeng@usq.edu.au (X.Z.); peter.schubel@usq.edu.au (P.S.); 2Wagners Composite Fibre Technologies, Wellcamp, QLD 4350, Australia; kendric.rendle-short@wagner.com.au; 3Allnex Composites, Wacol, QLD 4076, Australia; dean.voice@allnex.com

**Keywords:** pultrusion, DEA, cure behaviour, exothermic reaction, permittivity and ion viscosity, mechanical and thermal properties

## Abstract

Pultrusion is a high-volume manufacturing process for Fibre-Reinforced Polymer (FRP) composites. It requires careful tuning and optimisation of process parameters to obtain the maximum production rate. The present work focuses on the correlation between the set die temperatures of 80 °C, 100 °C, 120 °C, and 140 °C and the resin cure state at constant pull speeds. Lab-scale oven trials were conducted to understand the thermal behaviour of the resin system and to provide a temperature range for the pultrusion trials. Dielectric Analysis (DEA) was used during pultrusion trials to monitor the effect of die temperature on the cure progression. The DEA results showed that, by increasing die temperature, the exothermic peak shifts closer towards the die entry. Moreover, the degree of cure for samples processed at 140 °C was 97.7%, in comparison to 86.2% for those cured at 100 °C. The rate of conversion and the degree of cure correspond directly to the set die temperatures of the pultrusion trials, contributing to understanding the effect of die temperature on cure progression. Mechanical and thermal material properties were measured. Samples cured at 120 °C showed the highest mechanical performance, exceeding those cured at 140 °C, linked to the generation of higher internal stress due to the higher rate of conversion. This work can be used as a guide for pultruded composite sections, to understand the cure behaviour of resin systems under various applied temperatures and the impact of the die temperature conditions on thermal and mechanical properties.

## 1. Introduction

Fibre-Reinforced Polymers (FRPs) can deliver unique solutions for modern-day applications with cost-effectiveness, reliability, and compatibility [1]. There are various manufacturing techniques available for composite part designers and manufacturers, such as hand layup, spray layup, resin infusion, hot compression moulding, resin transfer moulding, filament winding, braiding, and pultrusion [2,3]. Performance, part complexity, efficiency, accuracy, cost, product mass, resin type, and fibre type dictate the most suitable manufacturing technique according to the product application.

Pultrusion as a composite manufacturing technique offers the ability to produce high-performance and cost-effective products, with less production waste for constant cross-section parts [4,5,6,7,8]. In pultrusion, the resin type, pot life, fibre fraction, cure kinetics, viscosity, wettability, and manufacturing conditions (including ambient conditions) are important factors to determine the process parameters [9,10,11]. Pultrusion production rates depend on parameters such as die temperature and line speed, which effect the cure progression of the resin, including the rate of cure and final degree of cure of the resin [10,12,13,14,15]. Characterisation and optimisation of the cure are essential in modifying the process parameters for increased production rates.

Characterisation of resin systems for various manufacturing processes has previously been studied with the help of analytical models including cure kinetic models. Li et al. [16] modeled the degree of cure using an algorithm to relate the pultrusion die temperature with the uniformity of cure; it was observed that higher die temperatures resulted in a higher final degree of cure. Ziaee et al. [17] investigated the relationship between cure kinetics, cure temperature, microstructure formation, and mechanical properties using the copolymer composition equation, finding that initial cure temperatures under isothermal heating conditions influence the mechanical properties of vinyl ester resins. Russell et al. [18] studied the effect of temperature and resin composition on isothermal cure kinetics with the help of an autocatalytic equation showing an increase in cure conversion with increased die temperature. In a study by Baran et al. [13], a semiempirical autocatalytic model was developed to predict the evolution of the degree of cure using isothermal and dynamic scan data. Thus, an elastic modulus evolution was predicted from a cure hardening and thermal softening modulus model using a modified CHILE approach. Zhang et al. [19] determined the modulus of the matrix using the Kamal model to predict internal stresses, prevalent in thicker profiles due to higher temperature gradients whilst curing. In a recent study by de Cassia Costa Dias et al. [20], the thermal and curing behaviour of epoxy resin was successfully predicted by using phenomenological kinetic models with diffusion effects. Yuksel et al. [21] developed a cure-degree- and temperature-dependent viscosity model to characterise and understand the evolution of thermo–chemo–rheological and mechanical material properties.

Previous studies have aimed to understand the influence of die temperature on the curing and thermal behaviour of pultruded sections with the help of numerical models [9,22,23,24,25]. For instance, Kommu et al. [9] developed a numerical model using finite element and finite different techniques, finding that the simulation helped in designing the die geometry along with optimising the processing conditions such as cure behaviour, heat transfer, and resin impregnation. Sandberg et al. [22] developed a numerical framework using the arbitrary Lagrangian–Eulerian approach to predict the nonisothermal heat transfer and resin flow in a resin-injection pultrusion process, validated by real temperature measurements from an industrial pultrusion line. Test results helped in understanding the cure behaviour and phase change in the resin system. Han et al. [23] developed a numerical model coupled with a kinetic expression to predict the temperature profile and degree of cure through the pultrusion die. Smolnicki et al. [24] developed a 3D numerical model and analysed heat transfer from the heaters to the profile with this approach, allowing for the simulation of different heating conditions and cure cycles in a pultrusion process. Barkanov et al. [26] developed electromagnetic thermochemical finite element models by replacing conventional heating sources with microwave-assisted curing to enhance the effectiveness and productivity. Safonov et al. [27] conducted a numerical study of pultruded rods and successfully simulated temperature distribution and degree of polymerisation. Diniz et al. [28] developed a numerical model through parametric analysis and predicted the strength of the pultruded profiles.

Experimental studies examining the kinetics of free radical polymerisation to investigate the rate and degree of cure of the resin through the curing die have previously been presented. Cebrian et al. [29] investigated the dual-step cure process of a paste adhesive, in which results showed that the curing time of the paste adhesive could be reduced from 4 h to 30 min by increasing the curing temperature of both curing steps and achieving the target degree of cure of 95%. This is due to the faster propagation and higher efficiency of the initiator, as observed by Scott et al. [30], where the rise of cure temperature from −30 °C to 90 °C showed that the rate of photopolymerisation was increased tenfold. Vedernikov et al. [31] conducted pultrusion trials with four different vinyl ester resin formulations and found that higher curing temperatures shifted the polymerisation/exothermic peak away from the die exit. Cook et al. [32,33] observed, by increasing the isothermal cure temperature, a higher cure rate and greater levels of conversion were resulted, due to the termination of the initiator as the cross linking network developed. Young et al. [34] reiterated the effect of higher temperatures on the rate of conversion; in this case, the cause is the diffusion control where the reaction rate is similar to the movement of the initiators throughout the reaction. Mahadevaswamy et al. [35] investigated the effect of high temperatures at 30 °C, 60 °C, 120 °C, and 160 °C on electrical and mechanical properties and found that greater temperatures such as 160 °C result in lower tensile properties due to the weakening of chemical bonds and reduced bond strength between fibre and matrix. In a study by Zhang et al. [36], a finite element model considering temperature-dependent material properties using the Hashin failure criterion was developed to simulate the web crippling behaviour of I-shaped pultruded profiles and showed that ultimate loads were reduced with the elevated temperatures due to the compressive and in-plane shear strength reduction.

The literature shows that modifications to the die temperature affect the final degree of cure. Moreover, the literature sheds light on ways to accelerate the cure reaction in various thermoset resin systems and applications [9,17,31]. However, a gap exists in understanding methodologies to represent an ideal pultrusion process without compromising the mechanical and thermal properties with accelerated cure reactions in an industrial-scale pultrusion line. Moreover, methods to modify the applied temperature assisting pultrusion manufacturers to scale up their production rates via greater line speeds require further investigation. Previous studies do not present in situ resin cure monitoring through the pultrusion die to validate assumptions and models, a gap to report on a unique experimental investigation showing the influence of the die temperature on accelerating cure progression advantages and disadvantages on mechanical and thermal properties.

This study therefore aimed to investigate the curing behaviour of a vinyl ester resin system under isothermal curing conditions on a Pultrex Px250-3T pultrusion machine using a systematic approach. Isothermal differential scanning calorimetry (DSC) trials were conducted to understand the cure behaviour of the resin used for the pultrusion samples. The DSC trials complement oven trials used to measure the cure progression response to the set oven temperature. Accordingly, this study provides an understanding of the influence of various curing die set temperatures on the critical features of the resin cure kinetics within a curing die, using Dielectric Analysis (DEA) sensors coupled with thermocouples traces. Pultruded samples will be analysed for mechanical and thermal material properties and compared. The outcome of this study will allow for faster implementation of new resin systems, leading to accelerated production rates on an industrial scale by providing evidence-based research and development techniques.

## 2. Materials and Methods

### 2.1. Materials

Unidirectional Jushi glass fibre rovings (4400 tex) were used in this study as the glass fibre reinforcement. A Viapal 4838 vinyl ester resin system (Allnex Composites, Wacol, Australia) was adopted for this work, a commercially available product manufactured in Australia. The resin recipe contains initiators including Perkadox 16 [37] (Nouryon, Amsterdam, The Netherlands), TBPEH [38], and TBPB [39] (United Initiators GmbH, Pullach, Germany). In addition, Vinnepas B60 and an internal mould release agent INT-PUL-24 (Axel Plastics, Monroe, CT, USA) were utilised in the final resin mix. The flat bar product manufactured is 56 mm wide and 3.25 mm thick and contains 82% unidirectional glass fibres by weight, measured using ASTM D2584 [40].

### 2.2. Methodology

#### 2.2.1. Differential Scanning Calorimetry

The DSC samples were prepared by filling the aluminium pans with mixed resin of 10 mg. Two samples were prepared for the isothermal temperatures of 80 °C, 100 °C, 120 °C, 140 °C, 160 °C, 180 °C, 200 °C, and 220 °C. The cure behaviour of the resin system is measured using a TA Instruments DSC (New Castle, DE, USA).

#### 2.2.2. Oven Trials

The samples for the oven trials were prepared by placing the J-type thermocouple from TC Direct (Oakleigh, Australia) adjacent to a dielectric sensor 115L IDEX (Figure 1a) from NETZSCH-Gerätebau GmbH, Selb, Germany. The cure monitoring of the resin was recorded with the Dielectric Analyser Ionic 288 from NETZSCH allowing the dielectric properties and temperature profile of the resin to be measured (Figure 1b); a frequency of 1 Hz was used for the analysis of results. Oven trials were performed using an oven from Laboratory Equipment Pty Ltd., Marrickville, Australia, using 30 g of resin in an aluminium pan, as shown in Figure 1a. Oven temperatures were varied from 80 °C to 240 °C, with an increment of 20 °C for each trial.

#### 2.2.3. Pultrusion Trials

Pultruded profiles were manufactured using a Px250-3T (Pultrex, Manningtree, UK) pultrusion machine, as seen in Figure 2a, using a resin bath setup. Unidirectional rovings only (4400 tex) were used for the trials, with a total of 64 to achieve the targeted fibre volume fraction of 65%, with a constant line speed of 500 mm/min. The impregnated fibres are then pulled into a one-meter-long curing die (Figure 2b), with the heater blocks split into four equal-length zones, each able to be controlled independently of one another. However, for this study, all the four sections were set at the same temperature to provide the isothermal heating condition. Setpoint temperatures investigated include 80 °C, 100 °C, 120 °C, and 140 °C. Five-metre pultruded samples would be pulled for each temperature set.

An IDEX sensor was inserted prior to the impregnated fibres entering the curing die to measure the cure behaviour through the length of the die. The Ionic 288 captured the data and associated software (Proteus for Thermal Analyzers) from NETZSCH is used to analyse the measured data, focusing on various dielectric variables such as log ion viscosity and permittivity along with the temperature profile, according to ASTM D150 [41].

#### 2.2.4. FTIR Analysis

The chemical composition of the cured samples from different trials was tested using a Thermo-Nicolet FTIR iS50 (Thermo Fisher Scientific, Waltham, MA, USA), using the ATR module; samples of the pultruded profiles were prepared to ensure that correct mounting was achieved to avoid defects such as voids and matrix cracks. Each spectrum was recorded until the wavelength of 4000 cm^−1^ was attained.

#### 2.2.5. Mechanical Testing

The mechanical properties of the pultruded samples were obtained using a 100 kN uniaxial testing machine (MTS Systems). Five samples from the pultruded profiles produced at each temperature were cut for each mechanical test employed. Flexural (ASTM D7264 [42]), interlaminar shear (ASTM D2344 [43]), in-plane shear (ASTM D5379 [44]), and compression (ASTM D6641 [43]) tests were performed. These mechanical properties were selected as the test results are driven by resin performance rather than the fibre performance [45,46,47]. The glass transition temperature was found using a Dynamic Mechanical Analyser Q 800 (TA Instruments, New Castle, DE, USA), in accordance with ASTM D7028 [48].

## 3. Results

### 3.1. Thermal Behaviour of Resin in DSC Trials

Figure 3 presents the results of the heat flow response using DSC of the resin and focuses on the difference in evolution of heat flow between 80 °C and 140 °C and the similarity in the heat flow beyond 140 °C until 220 °C. There is a distinct change in the heat flow peak between 80 °C and 140 °C, with a slight downward heat flow reduction trend above 140 °C.

### 3.2. Thermal Behaviour of Resin in Oven Trials

Figure 4 shows the thermal response of the resin during curing for isothermal conditions from 80 to 220 °C within an oven. It shows that the time to reach peak exothermic temperature decreases with an increase in cure temperature. The peak exothermic temperature between 80 °C and 140 °C increases substantially, with no noticeable difference in the peak temperature above 140 °C.

### 3.3. Dynamic Thermal Behaviour

Figure 5 shows temperature profile and the dielectric response of the pultruded specimens cured at 80 °C, 100 °C, 120 °C, and 140 °C. When the impregnated fibres enter the die, the temperature starts to increase, the initiators at specific temperatures activate once reached, and then the first two stages of polymerisation (initiation and propagation) begin to take place. This leads to a rise in heat driven by the exothermic reaction, which can be seen as contributing to the peak in the temperature profile. It can be observed from Figure 5 that the rate of reaction is directly proportional to the curing die temperature until the polymerisation begins. Thus, Table 1 reports that the time required to start the reaction decreases with the increase in the die temperature.

The permittivity curve in Figure 5 monitors the curing behaviour captured by the DEA sensors. An increase in the permittivity indicates an increase in resin viscosity because of increased temperature driving cross-linking. Accordingly, the peak of the permittivity shows the start of the chemical reaction until going down to the flat line, indicating the end of the reaction (resin consolidation).

DEA data help to identify the start of the reaction using permittivity as an indicator (Figure 5), whereas the peak temperature is considered to correlate with the maximum rate of cure. Figure 5 shows that an 80 °C die temperature and line speed of 500 mm/min are insufficient to kick off the first initiator, with no permittivity curve able to be presented. The rest of the trials show a clear temperature peak in the die, with further increases to the die temperature. In addition, it can be observed in Figure 5 that the increase in die temperature results in higher heating rates at the same line speed, also shown in Table 1 for the permittivity and log ion viscosity results.

### 3.4. Poymerisation Mechanisms and FTIR

FTIR tests were performed to determine the effect of curing parameters on the chemical composition of the pultruded samples. It is evident from Figure 6 that the peaks have attained similar wavenumbers for the three samples cured at 100 °C, 120 °C, and 140 °C. This indicates that the increase of the heating die has no noticeable effect on the chemical composition of the finished product when cured between 100 and 140 °C. Absorption bands were characterised by identifying the spectra of the functional groups, which start from the wavenumber of 1500 cm^−1.^ The medium bands observed at 2800–3000 cm^−1^ are attributed to C-H stretching alkane groups. The bands at 1700–1800 cm^−1^ would be assigned to C=O stretching ester groups. Weak bands at 1575–1620 cm^−1^ were observed and assigned to C=C stretching cyclic alkane groups.

Generally, the free radical polymerisation of vinyl ester constitutes three successive steps: initiation, propagation, and termination [49]. Initiators present in the resin mixture are responsible for the formation of active centres which attract the free monomers to form a chain reaction. There would be two events in the initiation process: the first would be the formation of free radicals from the initiator molecule, and the second would be the reaction of free radicals with the styrene monomers of the resin system. The rate of the active centre formation would be dependent on and controlled by the first step, i.e., free radical formation. Whilst the initiation kicks off the reaction by interconnecting the monomers and creating a chain nexus, the propagation helps in multiple identical chain formations in the reaction. The chain formation would be continuous until it is terminated by inhibitors or stoppers.

Figure 7 shows that initiators are responsible for the free radical formation which further reacts with the double-bonded monomers. The chemical reaction generates heat in the environment to convert the double-bonded monomer into a single-bonded electron-seeking molecule from the rest of the double-bonded monomers [50]. Further, the addition of molecules continues to acquire the electron, resulting in the chain formation, and this process is known as propagation; it stops only when the chain formation is paired with another radical.

### 3.5. Mechanical and Thermal Properties

The mechanical tests (interlaminar shear, flexure, in-plane shear, and compression) were performed, with results shown in Table 2. Specimens cured with a set die temperature of 120 °C have exhibited the highest mechanical properties.

The specimens pultruded at 120 °C and 140 °C showed an increase in the flexural strength of 29.7% and 5.6%, respectively, compared to the specimens pultruded at 100 °C. The increment in die temperature from 100 °C to 120 °C and 140 °C has an increase in the ILSS strength by 26.5% and 5.6%, respectively, compared to the 100 °C specimens. The in-plane shear strength of 120 °C and 140 °C specimens was found to be 27.8% and 2.8% greater than that of 100 °C specimens. Moreover, the compression strength at 120 °C and 140 °C showed 29.7% and 8.1% increases, respectively, compared to 100 °C specimens. Table 3 shows the glass transition temperatures (Tg) of the specimens pultruded at die temperatures of 100 °C, 120 °C, and 140 °C; an increase of 1.1% and 3.6% by increasing the die temperature from 120 °C and 140 °C, respectively.

## 4. Discussion

### 4.1. Influence of Temperature Increase on Curing Characteristics

The peak exothermic temperature in Figure 8 shows the effect of die temperature increases ranging from 100 °C to 120 °C and 140 °C. The peak reaction shifts in position towards the early portions of the die by 17.3% and 30.7% for the 120 °C and 140 °C cures, respectively, compared to the 100 °C used as the baseline in this study. This finding was also observed by Vedernikov et al. [31]. The significant increase in peak exothermic temperatures with the increase in die temperatures is due to the activation of multiple initiators simultaneously, as observed in Figure 7. Excessive heat is generated during the simultaneous conversion of double-bonded monomers at numerous locations by the activation of more than one initiator at respective high die temperatures [50]. It can be concluded from subsequent sections that with the increase in die temperature, curing can be accelerated; this, in turn, allows higher line speeds, enhancing manufacturing productivity. On the other hand, if there is no necessity of the greater line speed, a shorter die could be utilised for accelerated cure with the help of higher die temperature after considering the optimal mechanical properties.

Permittivity in Figure 8 shows that curing initiation occurred at positions further from the die entry for a 100 °C cure. On the contrary, the higher die temperatures (120 °C and 140 °C) result in initiation in earlier portions due to the cumulative action of initiators at higher temperatures. It is also observed from Table 1 and Figure 8 that the start of the reaction which is permittivity peak is happening before the exothermic peak for all the trials from 100 °C to 140 °C, which explains the correlation between DEA data and the exothermic temperatures. The permittivity curves provide evidence of the need for higher die temperatures to accelerate the cure reaction rate, resulting in the peak rate occurring in earlier portions of the die. This, in reverse, could increase the pulling force due to the friction exerted between the solid pultruded section and the die surface, which encourages shortening the die length to reduce the pulling force.

It is evident from Figure 8 that the log ion viscosity trends are very similar to that of the permittivity, showing a quicker reaction rate the higher the die temperature. It is clear from Table 1 that the observations of the time to start the reaction and to attain the exothermic peaks follow similar trends of accelerating the cure with the increased die temperatures. It was observed that the values of reaction start and exothermic peak indicate the possibility of pulling the profile with greater speeds.

### 4.2. Influence of Static and Dynamic Heating Conditions on Curing Behaviour

The polymerisation process is responsible for a resultant temperature peak which rises in these trials as die temperature is increased [50]. The three initiators used in this resin system activate styrene monomers in the resin mix and form an active centre that supports the formation of cross-linking chains. Similarly, the identical chains develop with the help of active centres, resulting in an exothermic reaction. The initiators are activated at their respective initiation temperatures to assist in the continuation of chain formation by the step transfer method. The reason the peak temperature of the resin system does not increase beyond 140 °C is that chain formation termination reaches a limit at this die temperature. The same is evident in Figure 4, as there is no significant change in the peak of the temperature profile between 140 °C and 230 °C [50]. This behaviour is limited to the resin system used in this study; an alteration to the recipe used for the resin will result in different responses to the set die temperatures.

The cure behaviour of the resin in the oven trials (static) and the pultrusion trials (dynamic) show similar trends (see Figure 9). Thus, an increase in the exothermic temperature can be observed until reaching 140 °C in both sets’ trials (see Figure 9a,b). This behaviour is also reported in the DSC results, as discussed in Section 3.1. Even with the same increase in the cure temperature, Figure 9b shows that there is a constant difference (around 50 °C) in the respective temperature peaks of the cure profile between oven and pultrusion trials. The reason for the lesser temperatures in the pultrusion trials compared to oven trials is attributed to the heat absorbed by the walls of the die acting as a heat sink [22]. For the resin system used in this study, 140 °C is considered the maximum pultrusion die temperature to be used, with no significant advantages gained by further increasing the die temperature. It supports the theory of identifying a ceasing temperature beyond which the cure is not affected by the incremental temperatures. In addition, it explains the phenomenon of having a specific temperature range for this resin to accelerate the cure.

### 4.3. Influence of Die Temperature on Mechanical and Thermal Properties

Die temperature has a direct influence on the mechanical properties of the pultruded specimens, as observed in Table 2. The 120 °C specimens attained the highest ultimate strengths, with the lowest observed for the 100 °C specimens. This is because the higher the temperature, the higher the propagation of cure, resulting in higher degree of cure and greater mechanical performance [17]. However, specimens pultruded at 140 °C show lower ultimate strengths than those at 120 °C. This can be attributed to the development of residual thermal stresses, as highlighted by Ziaee and Palmese [17]. The rapid increase in the curing rate causes improper structure formation in cross-linking during polymerisation. This can be validated by steep slope in the permittivity curve for the 140 °C trial compared to 120 °C in Figure 5, indicating the increase in the cure reaction rate. Thus, an optimum temperature range should be decided to keep up the mechanical properties and compromise accelerated cure.

The glass transition temperature shows a small increase with increased die temperature. An increase of 1.0% and 3.6% in the glass transition temperature was observed for the samples cured at 120 °C and 140 °C, respectively, compared to the 100 °C samples. The trend in the glass transition temperature does not follow that of the strength in this study; it shows that a higher Tg does not necessarily result in higher mechanical performance. Other factors influence the mechanical performance such as thermal stresses already noted in this study [17].

### 4.4. Cure Kinetic Predictions

Figure 10 shows the temperature profile, conversion progression, and rate of conversion of the resin system for each die temperature condition (80 °C, 100 °C, 120 °C, and 140 °C). It is assumed that the rate of the degree of cure (dα/dt) is proportional to the rate of heat flow (dH/dt), as seen in Equation (1).
(1)$$\frac{d\alpha }{dt}=\frac{1}{H_{tr}}\frac{dH}{dt}$$

An Arrhenius-type autocatalytic model is employed to predict the degree of cure and the rate of cure [31]:(2)$$R_{\gamma }\left(\alpha ,\;T\right)=\frac{d\alpha }{dt}=A_{0}\exp\left(\frac{-E_{a}}{RT}\right)\alpha^{m}{\left(1-\alpha \right)}^{n}$$
where *H_tr_* is the total exothermic heat of the reaction, *A*_0_ is the pre-exponential constant, *T* is the absolute temperature, *E_a_* is the activation energy, *R* is the universal gas constant, and *m* and *n* are the order of reaction. Equation (2) successfully predicted the conversion and rate of conversion, as seen in Figure 10.

For the 80 °C die temperature, there is no notable curing resulting in a solidified product; it is evident with a conversion rate of 6.9%/min and no visible peak in the rate of cure. In addition, the final degree of cure was 8.3%.

The cure behaviour for the 100 °C die temperature has shown that the maximum rate of cure was 200.6%/min and took 64 s from die entry, and the degree of cure was predicted to be 79.7% at the end of the die, and the final cure would be 86.2%.

For a 120 °C die temperature, the cure rate increased to 590.8%/min, which is more than twice the cure rate of the 100 °C die temperature, and resulted in 48 s. The degree of cure was 92.3% at the end of the die and 93.9% as the final conversion.

The die temperature of 140 °C resulted in the highest cure rate of 1102.6%/min, and the peak was attained at the quickest of all other temperatures, which is at 41 s. The degree of cure attained a sharp increase and attained 97.1% at the exit of the die, and the final cure was 97.7%, which turned out to be the greatest among the four die temperatures utilised in this study.

The analytical results using the cure kinetics model help in understanding the evolution of rate of conversion and the propagation of degree of cure relative to the various temperature profiles of the pultrusion trials. It is clear that the increment in die temperature results in accelerating the cure and shifting the reaction of cure to the earlier positions, as seen in Table 4, and which is also reflected in the dielectric parameters of Figure 8. It was observed that the accelerated cure resulted from the rise in conversion rate, as seen in Table 4. It was noticed that the degree of cure enhances with the rise in die temperatures and can be seen in the consistent increase in the final degree of cure simulations.

## 5. Conclusions

This paper presents a parametric study of the curing behaviour of a vinyl ester resin system in a pultrusion process by examining the influence of die temperature on the cure response at constant line speed. This work also presents a simplified methodology to understand the curing behaviour of a resin system using DSC and oven trials to identify an appropriate starting temperature range, prior to undertaking full-scale pultrusion trials. Proposing these simplified procedures assists in identifying the critical events in the curing process including the start of the reaction, end of the reaction, and the exothermic peak with respect to various die temperatures. Furthermore, the input of the temperature profiles from the pultrusion trials into a kinetic model helps to understand the cure behaviour and to predict the conversion rate and the final degree of cure. Accordingly, the following conclusions can be drawn:Oven trials and DSC tests are shown to be essential to understanding the curing behaviour of the resin by employing a systematic approach prior to its direct use in pultrusion, avoiding wasted resources if using a trial-and-error approach.Based on the resin system, this study shows that the oven-curing trials helped in understanding the curing rate and peak temperature event during curing under various isothermal temperatures. This allows an informed approach to be taken to initial pultrusion trials for selecting an initial die temperature range that can be used for the heating die in the pultrusion process with a prior prediction on the location of the curing within the heating die.With the aid of DEA sensors and thermocouples, the pultrusion trials show how an increase in die temperature from 80 °C to 140 °C results in an increase in the conversation rate, shifting the peak curing temperature towards the die entry. This behaviour was a result of rapid conversion of the double bonds during polymerisation. This finding can lead to further optimisation in the line speed or a reduction in the curing die length, both of which can positively contribute to the industrial productivity considering the optimal mechanical properties.The pultruded samples cured at 120 °C for this resin system show the greatest mechanical performance. The increased conversation rate of the 120 °C samples provided an increase in performance over the 100 °C samples (not rapid enough), with the conversion rate hindering the properties of the 140 °C samples (too rapid). The decrease in mechanical properties at high set temperatures could be attributed to the development of internal thermal stress due to the high exothermic temperatures. The DMA test results support the claim that the increase in die temperature increases the glass transition temperature.The analytical results using the cure kinetics model helped in understanding the development of the conversion at certain locations in the heating die. They also provide a clear demonstration of the curing rate at higher die temperature to some extent avoiding very rapid curing with sharp curing rate, as seen for samples cured at 140 °C. This supports the assertion that, in the case of excessive rapid curing, despite having a greater degree of cure, it can be to the detriment of the mechanical performance.

This work shows the effect of the die temperature on cure behaviour and mechanical and thermal properties. In addition, this methodology helps to understand the effective temperature range of the resin system with the help of oven trails and DSC tests even before conducting actual pultrusion runs. This methodology is recommended to be adapted for any resin system to understand its curing behaviour with respect to applied temperature range. Based on the outcome of this study, it is recommended further to investigate the effect of the production line speed on the curing behaviour of the resin system in order to achieve a higher production line speed.

## Figures and Tables

**Figure 1 polymers-15-03808-f001:**
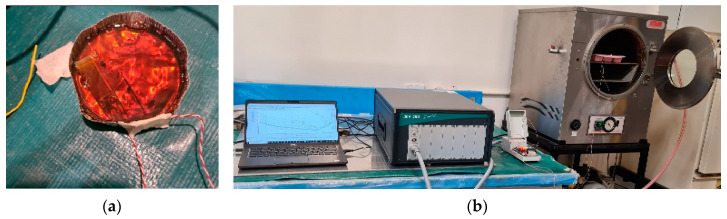
(**a**) Aluminium pan with thermocouple and DEA sensor in the resin. (**b**) Vacuum oven used in the lab-scale tests along with the DEA setup.

**Figure 2 polymers-15-03808-f002:**
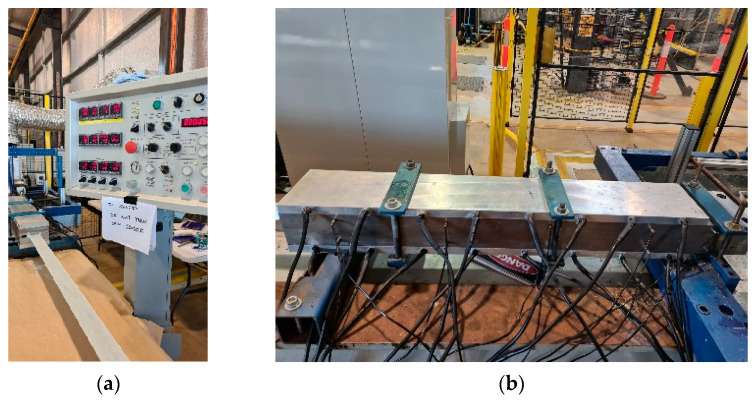
(**a**) Pultrusion setup at P11 laboratory, USQ Toowoomba. (**b**) Heated die used in the pultrusion trials.

**Figure 3 polymers-15-03808-f003:**
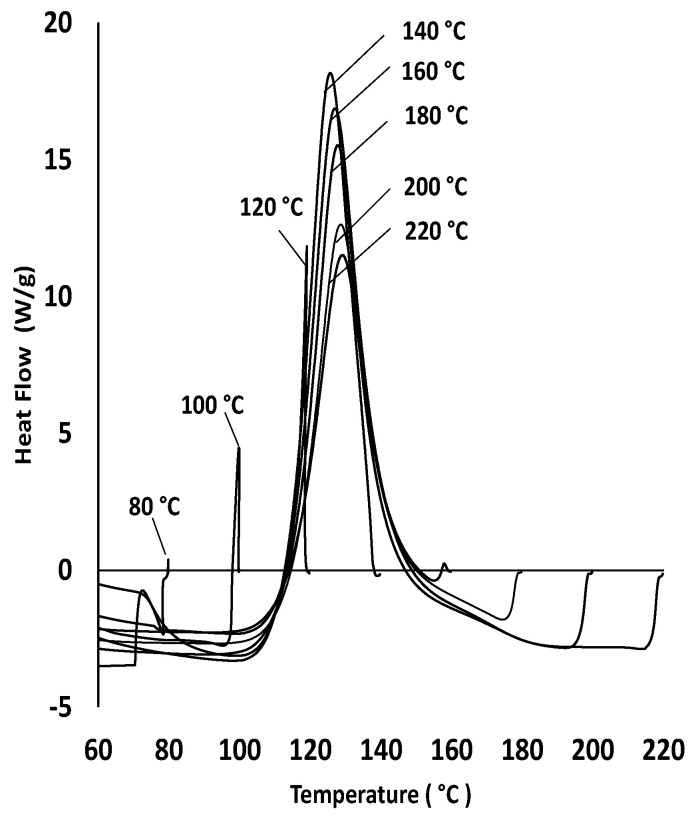
Heat flow behaviour of resin in DSC trials.

**Figure 4 polymers-15-03808-f004:**
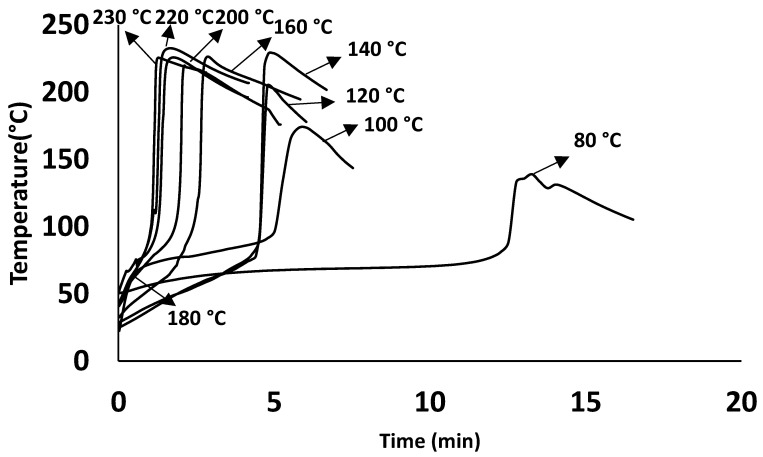
Time-dependent material behaviour under isothermal oven trials.

**Figure 5 polymers-15-03808-f005:**
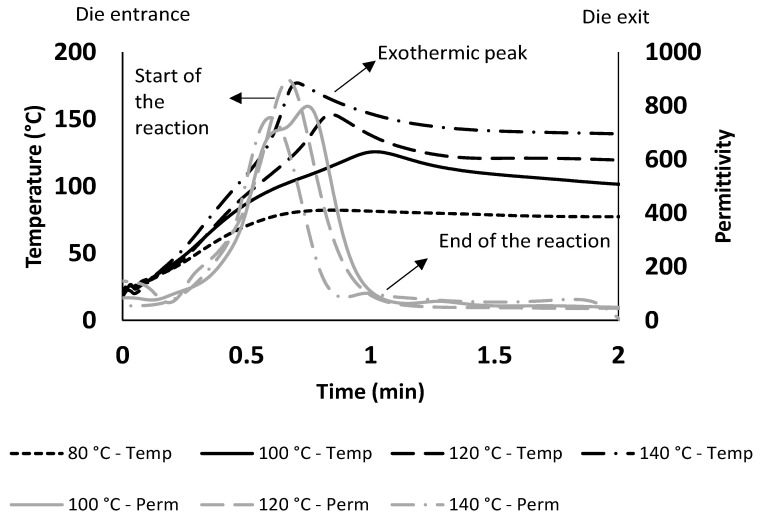
Temperature profile with DEA data of permittivity for pultruded profile at die temperatures of 100 °C, 120 °C, and 140 °C and line speed of 500 mm/min.

**Figure 6 polymers-15-03808-f006:**
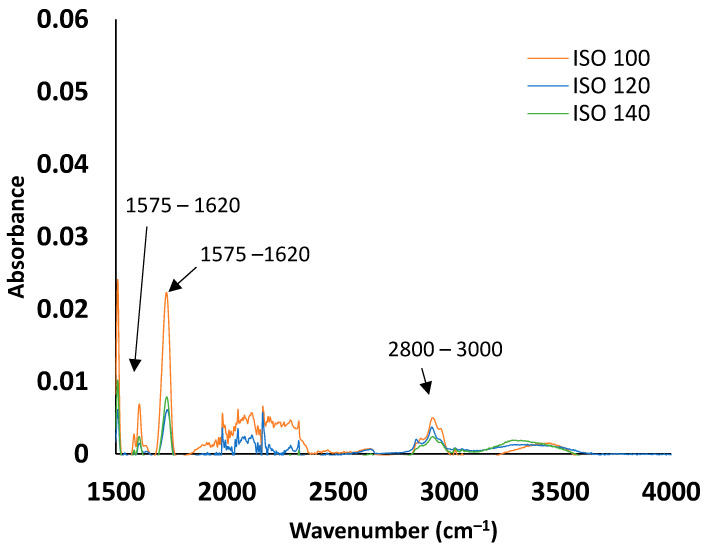
FTIR spectra of samples from the trials of die temperatures 100 °C, 120 °C, and 140 °C.

**Figure 7 polymers-15-03808-f007:**
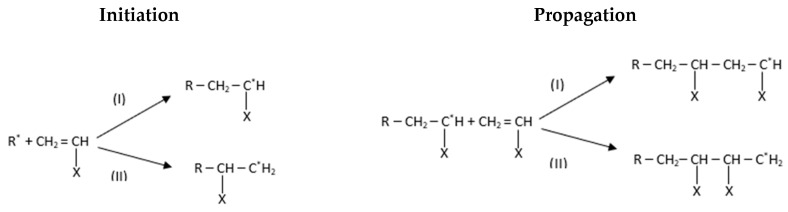
The chemical representation of chain reaction in the polymerisation process [50].

**Figure 8 polymers-15-03808-f008:**
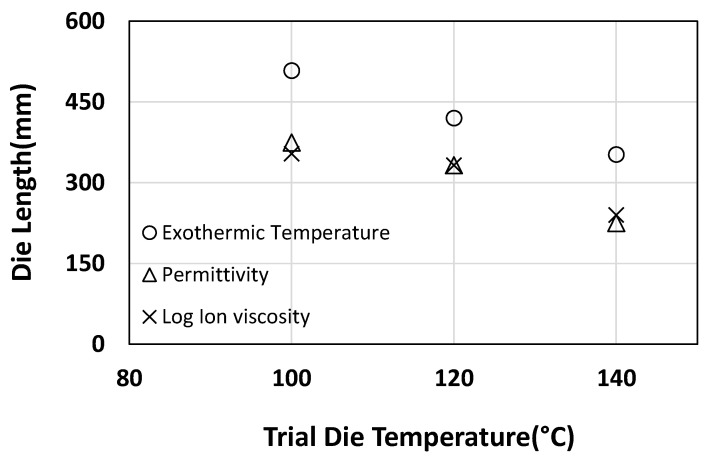
Curing critical points observed through the pultrusion trials at the die temperatures of 80 °C, 100 °C, 120 °C, and 140 °C.

**Figure 9 polymers-15-03808-f009:**
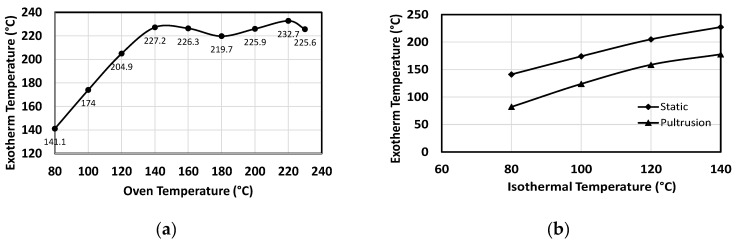
(**a**) Correlating exothermic temperature of resin and set oven temperature. (**b**) Correlating the exothermic temperatures of oven trials and pultrusion trials.

**Figure 10 polymers-15-03808-f010:**
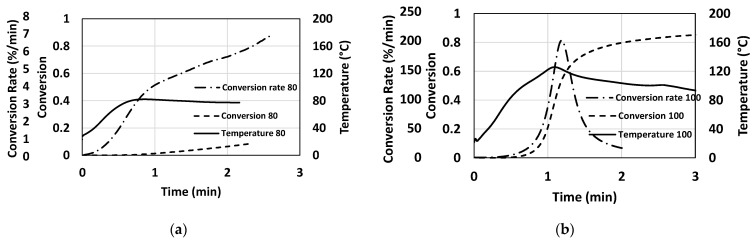
Prediction of conversion and rate of conversion corresponding to the temperature profiles of the pultrusion trails at the die temperatures (**a**) 80 °C, (**b**) 100 °C, (**c**) 120 °C, and (**d**) 140 °C.

**Table 1 polymers-15-03808-t001:** Observations of the start of the reaction and time to reach exothermic peak at different die temperatures.

Die Temperature (°C)	Start of the Reaction (Permittivity) (min)	Start of the Reaction (Log Ion Viscosity) (min)	Exothermic Peak (min)
80	-	-	-
100	0.74	0.70	1.01
120	0.66	0.66	0.84
140	0.45	0.48	0.70

**Table 2 polymers-15-03808-t002:** Mechanical test results in flexure and interlaminar shear.

Die Temperature (°C)	Flexure			Interlaminar Shear			In-Plane Shear			Compression		
	Mean (MPa)	St.D (MPa)	CoV (%)	Mean (MPa)	St.D (MPa)	CoV (%)	Mean (MPa)	St.D (MPa)	CoV (%)	Mean (MPa)	St.D (MPa)	CoV (%)
100	1052.7	35.0	3.3	438.8	18.2	4.1	36.7	1.3	3.5	529.5	37.0	6.0
120	1331.4	32.4	2.4	547.0	44.8	8.1	46.7	2.7	5.9	686.7	64.2	9.3
140	1111.7	28.1	2.5	464.6	11.3	2.4	35.4	1.6	4.6	572.9	43.1	7.5

**Table 3 polymers-15-03808-t003:** Glass transition temperature (Tg) of tested specimens of pultruded trials.

**Die Temperatures (°C)**	100	120	140
**Tg (°C)**	166.1	167.9	172.1
**Standard Deviation (°C)**	0.3	0.2	0.3

**Table 4 polymers-15-03808-t004:** Prediction of the important parameters of the cure relative to the various die temperatures.

Die Temperature (°C)	Conversion Rate (%/min)	Conversion Rate Peak (min)	Conversion at Die Exit (%)	Final Conversion (%)
80	6.9	-	7.2	8.3
100	200.6	1.07	79.7	86.2
120	590.8	0.80	92.3	93.9
140	1102.6	0.68	97.1	97.7

## Data Availability

The data required have been reported in this manuscript.

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
