# Peer review of "Characterisation of Curing of Vinyl Ester Resin in an Industrial Pultrusion Process: Influence of Die Temperature"

_polymers, 2023, doi:10.3390/polym15183808_

Round 1

Reviewer 1 Report

After a comprehensive review of the manuscript " Characterisation of Curing of Vinyl Ester Resin in an Industrial 2 Pultrusion Process: Influence of Die Temperature", Topic and paper are interesting. There is no doubt in the originality of the work. I'll try to clear my point through the following comments:

1.      In the abstract, you must give results of failure modes, load-carrying capacities, energy dissipation capacities, stiffness. etc., % is needed.

2.      The introduction of the manuscript is not well prepared. A more general literature review is needed, and the novelty of the research should be stated clearly. The importance of the presented work is not clarified. The novelty and importance of the research should be clarified using up-to-date research around the globe. The differences from previous studies must be specified clearly. Experiences from all over the world could help authors perform comprehensive research.

3.      It is recommended to enhance the literature review with recent pultrusion.

4.      You must also add a flow chart about your study.

5.      There are typos in the text and figures. Needs to be corrected.

6.      Please provide more details about the standard for testing. What is the limit for stopping the loading process?

7.      Please, define your boundary conditions for experimental analysis.

8.      What is the criterion of the authors for choosing the tested specimens? Using different specimens or even different scales could jeopardize the results of the research or not.

9.      The conclusion part is only a summary of the results. We need technical and general advice here, which others can use (both researchers and engineers). It would be more appropriate if the conclusions were based on the corrosion results.

10.  Which are the lessons learned? The authors have to clarify before acceptance.

Moderate editing of English language required

Author Response

Thanks to the respected reviewer for her/his comments, very informative and surely will contribute to add a value to the manuscript. The authors attempted to address the reviewer's comments as required in the attached file.

The authors hope that they have a drressed all the reviewer's enquiries.

Reviewer 2 Report

The article is useful to the industry in identifying the die temperature for proper curing and reduced time to increase the productivity. While the methods are well described, minor modifications are required to benefit the reader.

1) Authors need to provide information on the frequency used for the DEA analysis, if they had performed any frequency sweep as the permittivity and ion viscosity are dependent on the frequency of the applied electric field.

2) Given the myriad of mechanical tests performed to characterize the response with different die temperatures, did the authors look at the micro-structure of the cured part to identify and voids/defects due to accelerated cure

3) Were the long run performance of these parts evaluated as a part of this research to understand the durability of these structures, as fatigue behavior is very much dependent on the presence of any manufacturing defects that may arise due to accelerated curing times, if not consider these studies in future work.

Author Response

(The authors gave the same response as above.)

Round 2

Reviewer 1 Report

It is recommended to enhance the literature review with recent pultrusion. The article may be acceptable.

Minor editing of English language required

Author Response

The authors have attempted to enahnce the literature with recent available studies on pultrusion. Moreover, the whole manuscript was curefully proofread by the authors.
